# Combinatorial semi-bandit with known covariance

**Rémy Degenne**
LMPA, Université Paris Diderot
CMLA, ENS Paris-Saclay
degenne@cmla.ens-cachan.fr

**Vianney Perchet**
CMLA, ENS Paris-Saclay
CRITEO Research, Paris
perchet@normalesup.org

## Abstract

The combinatorial stochastic semi-bandit problem is an extension of the classical multi-armed bandit problem in which an algorithm pulls more than one arm at each stage and the rewards of all pulled arms are revealed. One difference with the single arm variant is that the dependency structure of the arms is crucial. Previous works on this setting either used a worst-case approach or imposed independence of the arms. We introduce a way to quantify the dependency structure of the problem and design an algorithm that adapts to it. The algorithm is based on linear regression and the analysis develops techniques from the linear bandit literature. By comparing its performance to a new lower bound, we prove that it is optimal, up to a poly-logarithmic factor in the number of pulled arms.

## 1 Introduction and setting

The multi-armed bandit problem (MAB) is a sequential learning task in which an algorithm takes at each stage a decision (or, "pulls an arm"). It then gets a reward from this choice, with the goal of maximizing the cumulative reward [Robbins, 1985]. We consider here its stochastic combinatorial extension, in which the algorithm chooses at each stage a subset of arms [Audibert et al., 2013, Cesa-Bianchi and Lugosi, 2012, Chen et al., 2013, Gai et al., 2012]. These arms could form, for example, the path from an origin to a destination in a network. In the combinatorial setting, contrary to the the classical MAB, the inter-dependencies between the arms can play a role (we consider that the distribution of rewards is invariant with time). We investigate here how the covariance structure of the arms affects the difficulty of the learning task and whether it is possible to design a unique algorithm capable of performing optimally in all cases from the simple scenario with independent rewards to the more challenging scenario of general correlated rewards.

Formally, at each stage $t \in \mathbb{N}, t \geq 1$, an algorithm pulls $m \geq 1$ arms among $d \geq m$. Such a set of $m$ arms is called an "action" and will be denoted by $A_t \in \{0,1\}^d$, a vector with exactly $m$ non-zero entries. The possible actions are restricted to an arbitrary fixed subset $\mathcal{A} \subset \{0,1\}^d$. After choosing action $A_t$, the algorithm receives the reward $A_t^\top X_t$, where $X_t \in \mathbb{R}^d$ is the vector encapsulating the reward of the $d$ arms at stage $t$. The successive reward vectors $(X_t)_{t \geq 1}$ are i.i.d with unknown mean $\mu \in \mathbb{R}^d$. We consider a semi-bandit feedback system: after choosing the action $A_t$, the algorithm observes the reward of each of the arms in that action, but not the other rewards. Other possible feedbacks previously studied include bandit (only $A_t^\top X_t$ is revealed) and full information ($X_t$ is revealed). The goal of the algorithm is to maximize the cumulated reward up to stage $T \geq 1$ or equivalently to minimize the expected regret, which is the difference of the reward that would have been gained by choosing the best action in hindsight $A^*$ and what was actually gained:

$$\mathbb{E}R_T = \mathbb{E}\sum_{t=1}^{T}(A^{*\top}\mu - A_t^\top \mu)\,.$$

For an action $A \in \mathcal{A}$, the difference $\Delta_A = (A^{*\top}\mu - A^{\top}\mu)$ is called gap of $A$. We denote by $\Delta_t$ the gap of $A_t$, so that regret rewrites as $\mathbb{E}R_T = \mathbb{E}\sum_{t=1}^{T}\Delta_t$. We also define the minimal gap of an arm, $\Delta_{i,\min} = \min_{\{A \in \mathcal{A}: i \in A\}} \Delta_A$.

This setting was already studied Cesa-Bianchi and Lugosi [2012], most recently in Combes et al. [2015], Kveton et al. [2015], where two different algorithms are used to tackle on one hand the case where the arms have independent rewards and on the other hand the general bounded case. The regret guaranties of the two algorithms are different and reflect that the independent case is easier. Another algorithm for the independent arms case based on Thompson Sampling was introduced in Komiyama et al. [2015]. One of the main objectives of this paper is to design a unique algorithm that can adapt to the covariance structure of the problem when prior information is available.

The following notations will be used throughout the paper: given a matrix $M$ (resp. vector $v$), its $(i,j)^{\text{th}}$ (resp. $i^{\text{th}}$) coefficient is denoted by $M^{(ij)}$ (resp. $v^{(i)}$). For a matrix $M$, the diagonal matrix with same diagonal as $M$ is denoted by $\Sigma_M$.

We denote by $\eta_t$ the noise in the reward, i.e. $\eta_t := X_t - \mu$. We consider a subgaussian setting, in which we suppose that there is a positive semi-definite matrix $C$ such that for all $t \geq 1$,

$$\forall u \in \mathbb{R}^d, \mathbb{E}[e^{u^{\top}\eta_t}] \leq e^{\frac{1}{2}u^{\top}Cu} .$$

This is equivalent to the usual setting for bandits where we suppose that the individual arms are subgaussian. Indeed if we have such a matrix $C$ then each $\eta_t^{(i)}$ is $\sqrt{C^{(ii)}}$-subgaussian. And under a subgaussian arms assumption, such a matrix always exists. This setting encompasses the case of bounded rewards.

We call $C$ a subgaussian covariance matrix of the noise (see appendix A of the supplementary material). A good knowledge of $C$ can simplify the problem greatly, as we will show. In the case of 1-subgaussian independent rewards, in which $C$ can be chosen diagonal, a known lower bound on the regret appearing in Combes et al. [2015] is $\frac{d}{\Delta}\log T$, while Kveton et al. [2015] proves a $\frac{dm}{\Delta}\log T$ lower bound in general. Our goal here is to investigate the spectrum of intermediate cases between these two settings, from the uninformed general case to the independent case in which one has much information on the relations between the arm rewards. We characterize the difficulty of the problem as a function of the subgaussian covariance matrix $C$. We suppose that we know a positive semi-definite matrix $\Gamma$ such that for all vectors $v$ with positive coordinates, $v^{\top}Cv \leq v^{\top}\Gamma v$, property that we denote by $C \preceq_+ \Gamma$. $\Gamma$ reflects the prior information available about the possible degree of independence of the arms. We will study algorithms that enjoy regret bounds as functions of $\Gamma$.

The matrix $\Gamma$ can be chosen such that all its coefficients are non-negative and verify for all $i,j$, $\Gamma^{(ij)} \leq \sqrt{\Gamma^{(ii)}\Gamma^{(jj)}}$. From now on, we suppose that it is the case. In the following, we will use $\epsilon_t$ such that $\eta_t = C^{1/2}\epsilon_t$ and write for the reward: $X_t = \mu + C^{1/2}\epsilon_t$.

## 2 Lower bound

We first prove a lower bound on the regret of any algorithm, demonstrating the link between the subgaussian covariance matrix and the difficulty of the problem. It depends on the maximal off-diagonal correlation coefficient of the covariance matrix. This coefficient is $\gamma = \max_{\{(i,j) \in [d], i \neq j\}} \frac{C^{(ij)}}{\sqrt{C^{(ii)}C^{(jj)}}}$. The bound is valid for consistent algorithms [Lai and Robbins, 1985], for which the regret on any problem verifies $\mathbb{E}R_t = o(t^a)$ as $t \to +\infty$ for all $a > 0$.

**Theorem 1.** *Suppose to simplify that $d$ is a multiple of $m$. Then, for any $\Delta > 0$, for any consistent algorithm, there is a problem with gaps $\Delta$, $\sigma$-subgaussian arms and correlation coefficients smaller than $\gamma \in [0,1]$ on which the regret is such that*

$$\liminf_{t \to +\infty} \frac{\mathbb{E}R_t}{\log t} \geq (1 + \gamma(m-1))\frac{2\sigma^2(d-m)}{\Delta}$$

This bound is a consequence of the classical result of Lai and Robbins [1985] for multi-armed bandits, applied to the problem of choosing one among $d/m$ paths, each of which has $m$ different successive edges (Figure 1). The rewards in the same path are correlated but the paths are independent. A complete proof can be found in appendix B.1 of the supplementary material.

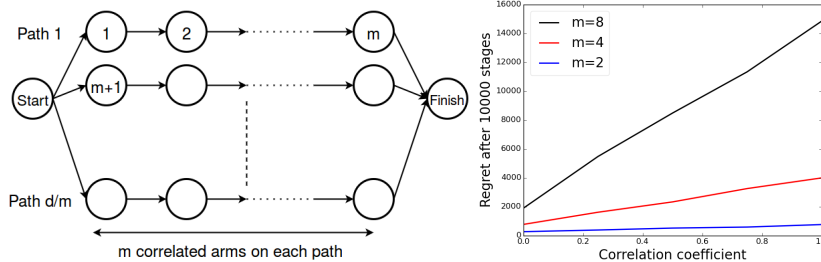

Figure 1: Left: parallel paths problem. Right: regret of OLS-UCB as a function of $m$ and $\gamma$ in the parallel paths problem with 5 paths (average over 1000 runs).

## 3 OLS-UCB Algorithm and analysis

Faced with the combinatorial semi-bandit at stage $t \geq 1$, the observations from $t-1$ stages form as many linear equations and the goal of an algorithm is to choose the best action. To find the action with the highest mean, we estimate the mean of all arms. This can be viewed as a regression problem. The design of our algorithm stems from this observation and is inspired by linear regression in the fixed design setting, similarly to what was done in the stochastic linear bandit literature [Rusmevichientong and Tsitsiklis, 2010, Filippi et al., 2010]. There are many estimators for linear regression and we focus on the one that is simple enough and adaptive: Ordinary Least Squares (OLS).

### 3.1 Fixed design linear regression and OLS-UCB algorithm

For an action $A \in \mathcal{A}$, let $I_A$ be the diagonal matrix with a 1 at line $i$ if $A^{(i)} = 1$ and 0 otherwise. For a matrix $M$, we also denote by $M_A$ the matrix $I_A M I_A$. At stage $t$, if all actions $A_1, \ldots, A_t$ were independent of the rewards, we would have observed a set of linear equations

$$I_{A_1} X_1 = I_{A_1} \mu + I_{A_1} \eta_1$$

$$\vdots$$

$$I_{A_{t-1}} X_{t-1} = I_{A_{t-1}} \mu + I_{A_{t-1}} \eta_{t-1}$$

and we could use the OLS estimator to estimate $\mu$, which is unbiased and has a known subgaussian constant controlling its variance. This is however not true in our online setting since the successive actions are not independent. At stage $t$, we define

$$n_t^{(i)} = \sum_{s=1}^{t-1} \mathbb{I}\{i \in A_s\}, \ n_t^{(ij)} = \sum_{s=1}^{t-1} \mathbb{I}\{i \in A_s\}\mathbb{I}\{j \in A_s\} \text{ and } D_t = \sum_{s=1}^{t-1} I_{A_s} \ ,$$

where $n_t^{(i)}$ is the number of times arm $i$ has been pulled before stage $t$ and $D_t$ is a diagonal matrix of these numbers. The OLS estimator is, for an arm $i \in [d]$,

$$\hat{\mu}_t^{(i)} = \frac{1}{n_t^{(i)}} \sum_{s < t : i \in A_s} X_s^{(i)} = \mu^{(i)} + \left( D_t^{-1} \sum_{s=1}^{t-1} I_{A_s} C^{1/2} \epsilon_s \right)^{(i)} .$$

Then for all $A \in \mathcal{A}$, $A^{\top}(\hat{\mu}_t - \mu)$ in the fixed design setting has a subgaussian matrix equal to $D_t^{-1} \left( \sum_{s=1}^{t-1} C_{A_s} \right) D_t^{-1}$. We get confidence intervals for the estimates and can use an upper confidence bound strategy [Lai and Robbins, 1985, Auer et al., 2002]. In the online learning setting the actions are not independent but we will show that using this estimator still leads to estimates that are well concentrated around $\mu$, with confidence intervals given by the same subgaussian matrix. The algorithm OLS-UCB (Algorithm 1) results from an application of an upper confidence bound strategy with this estimator.

We now turn to an analysis of the regret of OLS-UCB. At any stage $t \geq 1$ of the algorithm, let $\gamma_t = \max_{\{(i,j) \in A_t, i \neq j\}} \frac{\Gamma^{(ij)}}{\sqrt{\Gamma^{(ii)}\Gamma^{(jj)}}}$ be the maximal off-diagonal correlation coefficient of $\Gamma_{A_t}$ and let $\gamma = \max_{\{t \in [T]\}} \gamma_t$ be the maximum up to stage $T$.

---
**Algorithm 1** OLS-UCB.

---
**Require:** Positive semi-definite matrix $\Gamma$, real parameter $\lambda > 0$.
 1: Choose actions such that each arm is pulled at least one time.
 2: **loop**: at stage $t$,
 3:     $A_t = \arg\max_A A^\top \hat{\mu}_t + E_t(A)$
        with $E_t(A) = \sqrt{2f(t)}\sqrt{A^\top D_t^{-1}(\lambda \Sigma_\Gamma D_t + \sum_{s=1}^{t-1} \Gamma_{A_s})D_t^{-1}A}$.
 4:     Choose action $A_t$, observe $I_{A_t}X_t$.
 5:     Update $\hat{\mu}_t, D_t$.
 6: **end loop**

---

**Theorem 2.** *The OLS-UCB algorithm with parameter $\lambda > 0$ and $f(t) = \log t + (m+2)\log\log t + \frac{m}{2}\log(1 + \frac{e}{\lambda})$ enjoys for all times $T \geq 1$ the regret bound*

$$\mathbb{E}[R_T] \leq 16 f(T) \sum_{i \in [d]} \frac{\Gamma^{(ii)}}{\Delta_{i,\min}} \left( 5(\lambda + 1 - \gamma)\left\lceil \frac{\log m}{1.6} \right\rceil^2 + 45\gamma m \right)$$
$$+ \frac{8dm^2 \max_i\{C^{(ii)}\}\Delta_{\max}}{\Delta_{\min}^2} + 4\Delta_{\max},$$

*where $\lceil x \rceil$ stands for the smallest positive integer bigger than or equal to $x$. In particular, $\lceil 0 \rceil = 1$.*

This bound shows the transition between a general case with a $\frac{dm\log T}{\Delta}$ regime and an independent case with a $\frac{d\log^2 m \log T}{\Delta}$ upper bound (we recall that the lower bound is of the order of $\frac{d\log T}{\Delta}$). The weight of each case is given by the maximum correlation parameter $\gamma$. The parameter $\lambda$ seems to be an artefact of the analysis and can in practice be taken very small or even equal to 0.

Figure 1 illustrates the regret of OLS-UCB on the parallel paths problem used to derive the lower bound. It shows a linear dependency in $\gamma$ and supports the hypothesis that the true upper bound matches the lower bound with a dependency in $m$ and $\gamma$ of the form $(1 + \gamma(m-1))$.

**Corollary 1.** *The OLS-UCB algorithm with matrix $\Gamma$ and parameter $\lambda > 0$ has a regret bounded as*

$$\mathbb{E}[R_T] \leq \mathcal{O}\left(\sqrt{dT\log T \max_{i \in [d]}\{\Gamma^{(ii)}\} \left( 5(\lambda + 1 - \gamma)\left\lceil \frac{\log m}{1.6} \right\rceil^2 + 45\gamma m \right)}\right).$$

*Proof.* We write that the regret up to stage $T$ is bounded by $\Delta T$ for actions with gap smaller than some $\Delta$ and bounded using theorem 2 for other actions (with $\Delta_{\min} \geq \Delta$). Maximizing over $\Delta$ then gives the result. $\qquad\square$

### 3.2 Comparison with other algorithms

Previous works supposed that the rewards of the individual arms are in $[0,1]$, which gives them a $1/2$-subgaussian property. Hence we suppose $(\forall i \in [d], C^{(ii)} = 1/2)$ for our comparison.

In the independent case, our algorithm is the same as ESCB-2 from Combes et al. [2015], up to the parameter $\lambda$. That paper shows that ESCB-2 enjoys an $\mathcal{O}(\frac{d\sqrt{m}\log T}{\Delta})$ upper bound but our analysis tighten it to $\mathcal{O}(\frac{d\log^2 m \log T}{\Delta})$.

In the general (worst) case, Kveton et al. [2015] prove an $\mathcal{O}(\frac{dm\log T}{\Delta})$ upper bound (which is tight) using CombUCB1, a UCB based algorithm introduced in Chen et al. [2013] which at stage $t$ uses the exploration term $\sqrt{1.5\log t}\sum_{i \in A} 1/\sqrt{n_t^{(i)}}$. Our exploration term always verifies $E_t(A) \leq \sqrt{f(t)}\sum_{i \in A} 1/\sqrt{n_t^{(i)}}$ with $f(t) \approx \log t$ (see section 3.6). Their exploration term is a worst-case confidence interval for the means. Their broader confidence intervals however have the desirable property that one can find the action that realizes the maximum index by solving a linear optimization problem, making their algorithm computationally efficient, quality that both ESCB and OLS-UCB are lacking.

None of the two former algorithms benefits from guaranties in the other regime. The regret of ESCB in the general possibly correlated case is unknown and the regret bound for CombUCB1 is not improved in the independent case. In contrast, OLS-UCB is adaptive in the sense that its performance gets better when more information is available on the independence of the arms.

### 3.3 Regret Decomposition

Let $\mathbb{H}_{i,t} = \{|\hat{\mu}_t^{(i)} - \mu^{(i)}| \geq \frac{\Delta_t}{2m}\}$ and $\mathbb{H}_t = \cup_{i=1}^d \mathbb{H}_{i,t}$. $\mathbb{H}_t$ is the event that at least one coordinate of $\hat{\mu}_t$ is far from the true mean. Let $\mathbb{G}_t = \{A^{*\top}\mu \geq A^{*\top}\hat{\mu}_t + E_t(A^*)\}$ be the event that the estimate of the optimal action is below its true mean by a big margin. We decompose the regret according to these events:

$$R_T \leq \sum_{t=1}^T \Delta_t \mathbb{I}\{\overline{\mathbb{G}}_t, \overline{\mathbb{H}}_t\} + \sum_{t=1}^T \Delta_t \mathbb{I}\{\mathbb{G}_t\} + \sum_{t=1}^T \Delta_t \mathbb{I}\{\mathbb{H}_t\}$$

Events $\mathbb{G}_t$ and $\mathbb{H}_t$ are rare and lead to a finite regret (see below). We first simplify the regret due to $\overline{\mathbb{G}}_t \cap \overline{\mathbb{H}}_t$ and show that it is bounded by the "variance" term of the algorithm.

**Lemma 1.** *With the algorithm choosing at stage $t$ the action $A_t = \arg\max_A (A^\top \hat{\mu}_t + E_t(A))$, we have $\Delta_t \mathbb{I}\{\overline{\mathbb{G}}_t, \overline{\mathbb{H}}_t\} \leq 2E_t(A_t)\mathbb{I}\{\Delta_t \leq E_t(A_t)\}$.*

Proof in appendix B.2 of the supplementary material. Then the regret is cut into three terms,

$$R_T \leq 2\sum_{t=1}^T E_t(A_t)\mathbb{I}\{\Delta_t \leq 2E_t(A_t)\} + \sum_{t=1}^T \Delta_t \mathbb{I}\{\mathbb{G}_t\} + \sum_{t=1}^T \Delta_t \mathbb{I}\{\mathbb{H}_t\} .$$

The three terms will be bounded as follows:

- The $\mathbb{H}_t$ term leads to a finite regret from a simple application of Hoeffding's inequality.
- The $\mathbb{G}_t$ term leads to a finite regret for a good choice of $f(t)$. This is where we need to show that the exploration term of the algorithm gives a high probability upper confidence bound of the reward.
- The $E_t(A_t)$ term, or variance term, is the main source of the regret and is bounded using ideas similar to the ones used in existing works on semi-bandits.

### 3.4 Expected regret from $\mathbb{H}_t$

**Lemma 2.** *The expected regret due to the event $\mathbb{H}_t$ is $\mathbb{E}[\sum_{t=1}^T \Delta_t \mathbb{I}\{\mathbb{H}_t\}] \leq \frac{8dm^2 \max_i\{C^{(ii)}\}\Delta_{\max}}{\Delta_{\min}^2}$ .*

The proof uses Hoeffding's inequality on the arm mean estimates and can be found in appendix B.2 of the supplementary material.

### 3.5 Expected regret from $\mathbb{G}_t$

We want to bound the probability that the estimated reward for the optimal action is far from its mean. We show that it is sufficient to control a self-normalized sum and do it using arguments from Peña et al. [2008], or Abbasi-Yadkori et al. [2011] who applied them to linear bandits. The analysis also involves a peeling argument, as was done in one dimension by Garivier [2013] to bound a similar quantity.

**Lemma 3.** *Let $\delta_t > 0$. With $\tilde{f}(\delta_t) = \log(1/\delta_t) + m \log\log t + \frac{m}{2}\log(1 + \frac{e}{\lambda})$ and an algorithm given by the exploration term $E_t(A) = \sqrt{2\tilde{f}(\delta_t)}\sqrt{A^\top D_t^{-1}(\lambda\Sigma_\Gamma D_t + \sum_{s=1}^{t-1}\Gamma_{A_s})D_t^{-1}A}$ , then the event $\mathbb{G}_t = \{A^{*\top}\mu \geq A^{*\top}\hat{\mu}_t + E_t(A^*)\}$ verifies $\mathbb{P}\{\mathbb{G}_t\} \leq \delta_t$ .*

With $\delta_1 = 1$ and $\delta_t = \frac{1}{t\log^2 t}$ for $t \geq 2$, such that $\tilde{f}(\delta_t) = f(t)$, the regret due to $\mathbb{G}_t$ is finite in expectation, bounded by $4\Delta_{\max}$.

*Proof.* We use a peeling argument: let $\eta > 0$ and for $a = (a_1, \ldots, a_m) \in \mathbb{N}^m$, let $\mathcal{D}_a \subset [T]$ be a subset of indices defined by $(t \in \mathcal{D}_a \Leftrightarrow \forall i \in A^*, (1+\eta)^{a_i} \leq n_t^{(i)} < (1+\eta)^{a_i+1})$. For any $\mathcal{B}_t \in \mathbb{R}$,

$$\mathbb{P}\left\{A^{*\top}(\mu - \hat{\mu}_t) \geq \mathcal{B}_t\right\} \leq \sum_a \mathbb{P}\left\{A^{*\top}(\mu - \hat{\mu}_t) \geq \mathcal{B}_t | t \in \mathcal{D}_a\right\}.$$

The number of possible sets $\mathcal{D}_a$ for $t$ is bounded by $(\log t / \log(1+\eta))^m$, since each number of pulls $n_t^{(i)}$ for $i \in A^*$ is bounded by $t$. We now search a bound of the form $\mathbb{P}\left\{A^{*\top}(\mu - \hat{\mu}_t) \geq \mathcal{B}_t | t \in \mathcal{D}_a\right\}$. Suppose $t \in \mathcal{D}_a$ and let $D$ be a positive definite diagonal matrix (that depends on $a$).

Let $S_t = \sum_{s=1}^{t-1} I_{A_s \cap A^*} C^{1/2} \epsilon_s$, $V_t = \sum_{s=1}^{t-1} C_{A_s \cap A^*}$ and $I_{V_t+D}(\epsilon) = \frac{1}{2} \|S_t\|_{(V_t+D)^{-1}}^2$.

**Lemma 4.** *Let $\delta_t > 0$ and let $\tilde{f}(\delta_t)$ be a function of $\delta_t$. With a choice of $D$ such that $I_{A^*} D \preceq \lambda I_{A^*} \Sigma_C D_t$ for all $t$ in $\mathcal{D}_a$,*

$$\mathbb{P}\left\{A^{*\top}(\mu - \hat{\mu}_t) \geq \sqrt{2\tilde{f}(\delta_t) A^{*\top} D_t^{-1}(\lambda \Sigma_C D_t + V_t) D_t^{-1} A^*} \, \Big| t \in \mathcal{D}_a\right\} \leq \mathbb{P}\left\{I_{V_t+D}(\epsilon) \geq \tilde{f}(\delta_t) | t \in \mathcal{D}_a\right\}.$$

Proof in appendix B.2 of the supplementary material.

The self-normalized sum $I_{V_t}(\epsilon)$ is an interesting quantity for the following reason: $\exp(\frac{1}{2} I_{V_t}(\epsilon)) = \max_{u \in \mathbb{R}^d} \prod_{s=1}^{t-1} \exp(u^\top I_{A_s \cap A^*} C^{1/2} \epsilon_s - u^\top C_{A_s \cap A^*} u)$. For a given $u$, the exponential is smaller that 1 in expectation, from the subgaussian hypothesis. The maximum of the expectation is then smaller than 1. To control $I_{V_t}(\epsilon)$, we are however interested in the expectation of this maximum and cannot interchange max and $\mathbb{E}$. The method of mixtures circumvents this difficulty: it provides an approximation of the maximum by integrating the exponential against a multivariate normal centered at the point $V_t^{-1} S_t$, where the maximum is attained. The integrals over $u$ and $\epsilon$ can then be swapped by Fubini's theorem to get an approximation of the expectation of the maximum using an integral of the expectations. Doing so leads to the following lemma, extracted from the proof of Theorem 1 of Abbasi-Yadkori et al. [2011].

**Lemma 5.** *Let $D$ be a positive definite matrix that does not depend on $t$ and* $M_t(D) = \sqrt{\frac{\det D}{\det(V_t+D)}} \exp(I_{V_t+D}(\epsilon))$. *Then* $\mathbb{E}[M_t(D)] \leq 1$.

We rewrite $\mathbb{P}\left\{I_{V_t+D}(\epsilon) \geq \tilde{f}(\delta_t)\right\}$ to introduce $M_t(D)$,

$$\mathbb{P}\left\{I_{V_t+D}(\epsilon) \geq \tilde{f}(\delta_t) | t \in \mathcal{D}_a\right\} = \mathbb{P}\left\{M_t(D) \geq \frac{1}{\sqrt{\det(I_d + D^{-1/2} V_t D^{-1/2})}} \exp(\tilde{f}(\delta_t)) \Big| t \in \mathcal{D}_a\right\}.$$

The peeling lets us bound $V_t$. Let $D_a$ be the diagonal matrix with entry $(i,i)$ equal to $(1+\eta)^{a_i}$ for $i \in A^*$ and 0 elsewhere.

**Lemma 6.** *With* $D = \lambda \Sigma_C D_a + I_{[d] \backslash A^*}$, $\det(I_d + D^{-1/2} V_t D^{-1/2}) \leq (1 + \frac{1+\eta}{\lambda})^m$.

The union bound on the sets $\mathcal{D}_a$ and Markov's inequality give

$$\mathbb{P}\left\{A^{*\top}(\mu - \hat{\mu}_t) \geq \sqrt{2\tilde{f}(\delta_t)} \sqrt{\lambda A^{*\top} \Sigma_C D_t^{-1} A^* + A^{*\top} D_t^{-1} V_t D_t^{-1} A^*}\right\}$$

$$\leq \sum_{\mathcal{D}_a} \mathbb{P}\left\{M_t(D) \geq (1 + \frac{1+\eta}{\lambda})^{-m/2} \exp(\tilde{f}(\delta_t)) | t \in \mathcal{D}_a\right\}$$

$$\leq \left(\frac{\log t}{\log(1+\eta)}\right)^m (1 + \frac{1+\eta}{\lambda})^{m/2} \exp(-\tilde{f}(\delta_t))$$

For $\eta = e - 1$ and $\tilde{f}(\delta_t)$ as in lemma 3, this is bounded by $\delta_t$. The result with $\Gamma$ instead of $C$ is a consequence of $C \preceq_+ \Gamma$. With $\delta_1 = 1$ and $\delta_t = 1/(t \log^2 t)$ for $t \geq 2$, the regret due to $\mathbb{G}_t$ is

$$\mathbb{E}[\sum_{t=1}^T \Delta_t \mathbb{I}\{\mathbb{G}_t\}] \leq \Delta_{\max}(1 + \sum_{t=2}^T \frac{1}{t \log^2 t}) \leq 4\Delta_{\max}.$$

$\square$

### 3.6 Bounding the variance term

The goal of this section is to bound $E_t(A_t)$ under the event $\{\Delta_t \leq E_t(A_t)\}$. Let $\gamma_t \in [0,1]$ such that for all $i,j \in A_t$ with $i \neq j$, $\Gamma^{(ij)} \leq \gamma_t\sqrt{\Gamma^{(ii)}\Gamma^{(jj)}}$. From the Cauchy-Schwartz inequality, $n_t^{(ij)} \leq \sqrt{n_t^{(i)}n_t^{(j)}}$. Using these two inequalities,

$$A_t^\top D_t^{-1}(\sum_{s=1}^{t-1}\Gamma_{A_s})D_t^{-1}A_t = \sum_{i,j\in A_t}\frac{n_t^{(ij)}\Gamma^{(ij)}}{n_t^{(i)}n_t^{(j)}} \leq (1-\gamma_t)\sum_{i\in A_t}\frac{\Gamma^{(ii)}}{n_t^{(i)}} + \gamma_t(\sum_{i\in A_t}\sqrt{\frac{\Gamma^{(ii)}}{n_t^{(i)}}})^2 .$$

We recognize here the forms of the indexes used in Combes et al. [2015] for independent arms (left term) and Kveton et al. [2015] for general arms (right term). Using $\Delta_t \leq E_t(A_t)$ we get

$$\frac{\Delta_t^2}{8f(t)} \leq (\lambda+1-\gamma_t)\sum_{i\in A_t}\frac{\Gamma^{(ii)}}{n_t^{(i)}} + \gamma_t(\sum_{i\in A_t}\sqrt{\frac{\Gamma^{(ii)}}{n_t^{(i)}}})^2 . \tag{1}$$

The strategy from here is to find events that must happen when (1) holds and to show that these events cannot happen very often. For positive integers $j$ and $t$ and for $e \in \{1,2\}$, we define the set of arms in $A_t$ that were pulled less than a given threshold: $S_{t,e}^j = \{i \in A_t, n_t^{(i)} \leq \alpha_{j,e}\frac{8f(t)\Gamma^{(ii)}g_e(m,\gamma_t)}{\Delta_t^2}\}$, with $g_e(m,\gamma_t)$ to be stated later and $(\alpha_{i,e})_{i\geq 1}$ a decreasing sequence. Let also $S_{t,e}^0 = A_t$. $(S_{t,e}^j)_{j\geq 0}$ is decreasing for the inclusion of sets and we impose $\lim_{j\to+\infty}\alpha_{j,e} = 0$, such that there is an index $j_\emptyset$ with $S_{t,e}^{j_\emptyset} = \emptyset$. We introduce another positive sequence $(\beta_{j,e})_{j\geq 0}$ and consider the events that at least $m\beta_{j,e}$ arms in $A_t$ are in the set $S_{t,e}^j$ and that the same is false for $k < j$, i.e. for $t \geq 1$, $\mathbb{A}_{t,e}^j = \{|S_{t,e}^j| \geq m\beta_{j,e}; \forall k < j, |S_{t,e}^k| < m\beta_{k,e}\}$. To avoid having some of these events being impossible we choose $(\beta_{j,e})_{j\geq 0}$ decreasing. We also impose $\beta_{0,e} = 1$, such that $|S_{t,e}^0| = m\beta_{0,e}$.

Let then $\mathbb{A}_{t,e} = \cup_{j=1}^{+\infty}\mathbb{A}_{t,e}^j$ and $\mathbb{A}_t = \mathbb{A}_{t,1} \cup \mathbb{A}_{t,2}$. We will show that $\mathbb{A}_t$ must happen for (1) to be true. First, remark that under a condition on $(\beta_{j,e})_{j\geq 0}$, $\mathbb{A}_t$ is a finite union of events,

**Lemma 7.** *For $e \in \{1,2\}$, if there exists $j_{0,e}$ such that $\beta_{j_{0,e},e} \leq 1/m$, then $\mathbb{A}_{t,e} = \cup_{j=1}^{j_0}\mathbb{A}_{t,e}^j$.*

We now show that $\overline{\mathbb{A}_t}$ is impossible by proving a contradiction in (1).

**Lemma 8.** *Under the event $\overline{\mathbb{A}_{t,1}}$, if there exists $j_0$ such that $\beta_{j_0,1} \leq 1/m$, then*

$$\sum_{i\in A_t}\frac{\Gamma^{(ii)}}{n_t^{(i)}} < \frac{m\Delta_t^2}{8f(t)g_1(m,\gamma_t)}\left(\sum_{j=1}^{j_0}\frac{\beta_{j-1,1}-\beta_{j,1}}{\alpha_{j,1}} + \frac{\beta_{j_0,1}}{\alpha_{j_0,1}}\right) .$$

*Under the event $\overline{\mathbb{A}_{t,2}}$, if $\lim_{j\to+\infty}\beta_{j,2}/\sqrt{\alpha_{j,2}} = 0$ and $\sum_{j=1}^{+\infty}\frac{\beta_{j-1,2}-\beta_{j,2}}{\sqrt{\alpha_{j,2}}}$ exists, then*

$$\sum_{i\in A_t}\sqrt{\frac{\Gamma^{(ii)}}{n_t^{(i)}}} \leq \frac{m\Delta_t}{\sqrt{8f(t)g_2(m,\gamma_t)}}\sum_{j=1}^{+\infty}\frac{\beta_{j-1,2}-\beta_{j,2}}{\sqrt{\alpha_{j,2}}} .$$

A proof can be found in appendix B.2 of the supplementary material. To ensure that the conditions of these lemmas are fulfilled, we impose that $(\beta_{i,1})_{i\geq 0}$ and $(\beta_{i,2})_{i\geq 0}$ have limit 0 and that $\lim_{j\to+\infty}\beta_{j,2}/\sqrt{\alpha_{j,2}} = 0$. Let $j_{0,1}$ be the smallest integer such that $\overline{\beta}_{j_{0,1},1} \leq 1/m$. Let $l_1 = \frac{\beta_{j_{0,1},1}}{\alpha_{j_{0,1},1}} + \sum_{j=1}^{j_{0,1}}\frac{\beta_{j-1,1}-\beta_{j,1}}{\alpha_{j,1}}$ and $l_2 = \sum_{j=1}^{+\infty}\frac{\beta_{j-1,2}-\beta_{j,2}}{\sqrt{\alpha_{j,2}}}$. Using the two last lemmas with (1), we get that if $\overline{\mathbb{A}_t}$ is true,

$$\frac{\Delta_t^2}{8f(t)} < \frac{\Delta_t^2}{8f(t)}\left((\lambda+1-\gamma_t)\frac{ml_1}{g_1(m,\gamma_t)} + \gamma_t\frac{m^2l_2^2}{g_2(m,\gamma_t)}\right) .$$

Taking $g_1(m,\gamma_t) = 2(\lambda+1-\gamma_t)ml_1$ and $g_2(m,\gamma_t) = 2\gamma_t m^2l_2^2$, we get a contradiction. Hence with these choices $\mathbb{A}_t$ must happen. The regret bound will be obtained by a union bound on the events that form $\mathbb{A}_t$. First suppose that all gaps are equal to the same $\Delta$.

**Lemma 9.** *Let* $\gamma = \max_{t \geq 1} \gamma_t$. *For* $j \in \mathbb{N}^*$, *the event* $\mathbb{A}_{t,e}^j$ *happens at most* $\frac{d\alpha_{j,e} 8 f(T) \max_i \{\Gamma^{(ii)}\} g_e(m,\gamma)}{m\beta_{j,e}\Delta^2}$ *times.*

*Proof.* Each time that $\mathbb{A}_{t,e}^j$ happens, the counter of plays $n_t^{(i)}$ of at least $m\beta_{j_e}$ arms is incremented. After $\frac{\alpha_{j,e} 8 f(T) \max_i \{\Gamma^{(ii)}\} g_e(m,\gamma)}{\Delta^2}$ increments, an arm cannot verify the condition on $n_t^{(i)}$ any more. There are $d$ arms, so the event can happen at most $d\frac{1}{m\beta_{j_e}}\frac{\alpha_{j,e} 8 f(T) \max_i \{\Gamma^{(ii)}\} g_e(m,\gamma)}{\Delta^2}$ times. $\qquad\square$

If all gaps are equal to $\Delta$, an union bound for $\mathbb{A}_t$ gives

$$\mathbb{E}[\sum_{t=1}^{T}\Delta\mathbb{I}\{\overline{\mathbb{H}}_t \cap \overline{\mathbb{G}}_t\}] \leq 16\max_{i\in[d]}\{\Gamma^{(ii)}\}\frac{f(T)}{\Delta}d\left[(\lambda+1-\gamma)l_1\sum_{j=1}^{j_{0,1}}\frac{\alpha_{j,1}}{\beta_{j,1}}+\gamma m l_2^2\sum_{j=1}^{+\infty}\frac{\alpha_{j,2}}{\beta_{j,2}}\right].$$

The general case requires more involved manipulations but the result is similar and no new important idea is used. The following lemma is proved in appendix B.2 of the supplementary material:

**Lemma 10.** *Let* $\gamma^{(i)} = \max_{\{t, i\in A_t\}} \gamma_t$. *The regret from the event* $\overline{\mathbb{H}}_t \cap \overline{\mathbb{G}}_t$ *is such that*

$$\mathbb{E}[\sum_{t=1}^{T}\Delta_t\mathbb{I}\{\overline{\mathbb{H}}_t \cap \overline{\mathbb{G}}_t\}] \leq 16f(T)\sum_{i\in[d]}\frac{\Gamma^{(ii)}}{\Delta_{i,\min}}\left[(\lambda+1-\gamma)l_1\sum_{j=1}^{j_0}\frac{\alpha_{j,1}}{\beta_{j,1}}+\gamma m l_2^2\sum_{j=1}^{+\infty}\frac{\alpha_{j,2}}{\beta_{j,2}}\right].$$

Finally we can find sequences $(\alpha_{j,1})_{j\geq1}$, $(\alpha_{j,2})_{j\geq1}$, $(\beta_{j,1})_{j\geq0}$ and $(\beta_{j,2})_{j\geq0}$ such that

$$\mathbb{E}[\sum_{t=1}^{T}\Delta\mathbb{I}\{\overline{\mathbb{H}}_t \cap \overline{\mathbb{G}}_t\}] \leq 16f(T)\sum_{i\in[d]}\frac{\Gamma^{(ii)}}{\Delta_{i,\min}}\left(5(\lambda+1-\gamma^{(i)})\left\lceil\frac{\log m}{1.6}\right\rceil^2+45\gamma^{(i)}m\right)$$

See appendix C of the supplementary material. In Combes et al. [2015], $\alpha_{i,1}$ and $\beta_{i,1}$ were such that the $\log^2 m$ term was replaced by $\sqrt{m}$. Our choice is also applicable to their ESCB algorithm. Our use of geometric sequences is only optimal among sequences such that $\beta_{i,1} = \alpha_{i,1}$ for all $i \geq 1$. It is unknown to us if one can do better. With this control of the variance term, we finally proved Theorem 2.

## 4 Conclusion

We defined a continuum of settings from the general to the independent arms cases which is suitable for the analysis of semi-bandit algorithms. We exhibited a lower bound scaling with a parameter that quantifies the particular setting in this continuum and proposed an algorithm inspired from linear regression with an upper bound that matches the lower bound up to a $\log^2 m$ term. Finally we showed how to use tools from the linear bandits literature to analyse algorithms for the combinatorial bandit case that are based on linear regression.

It would be interesting to estimate the subgaussian covariance matrix online to attain good regret bounds without prior knowledge. Also, our algorithm is not computationally efficient since it requires the computation of an argmax over the actions at each stage. It may be possible to compute this argmax less often and still keep the regret guaranty, as was done in Abbasi-Yadkori et al. [2011] and Combes et al. [2015].

On a broader scope, the inspiration from linear regression could lead to algorithms using different estimators, adapted to the structure of the problem. For example, the weighted least-square estimator is also unbiased and has smaller variance than OLS. Or one could take advantage of a sparse covariance matrix by using sparse estimators, as was done in the linear bandit case in Carpentier and Munos [2012].

## Acknowledgements

The authors would like to acknowledge funding from the ANR under grant number ANR-13-JS01-0004 as well as the Fondation Mathématiques Jacques Hadamard and EDF through the Program Gaspard Monge for Optimization and the Irsdi project Tecolere.

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
