[Supplementary Material]

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

# A    The subgaussian covariance matrix

**Property 0.**    An $\alpha$-subgaussian variable $X$ verifies $\mathrm{Var}[X] \leq \alpha^2$.

Let $\eta$ be a noise in $\mathbb{R}^d$ with mean 0 and subgaussian covariance matrix $C$. The following properties are immediate consequences of property 0 and are presented as a justification for the name subgaussian "covariance" matrix.

**Property 1.**    For all $i \in [d]$, $\mathrm{Var}[\eta^{(i)}] \leq C^{(ii)}$.

**Property 2.**    For all $(i,j) \in [d]$, $\mathrm{Var}[\eta^{(i)}] + \mathrm{Var}[\eta^{(j)}] + 2\mathrm{Cov}[\eta^{(i)}, \eta^{(j)}] \leq C^{(ii)} + C^{(jj)} + 2C^{(ij)}$.

**Property 3.**    If $\mathrm{Var}[\eta^{(i)}] = C^{(ii)}$ and $\mathrm{Var}[\eta^{(j)}] = C^{(jj)}$, $\mathrm{Cov}[\eta^{(i)}, \eta^{(j)}] \leq C^{(ij)}$.

# B   Missing proofs

## B.1   Lower bound

*Proof.* We consider the problem where $\mathcal{A}$ is a set of $d/m$ disjoint actions $A_1, \ldots, A_{d/m}$ and suppose that these actions are independent. This is not more than a bandit problem with $d/m$ arms, each with $m$ sub-arms. We choose a noise $\epsilon \sim \mathcal{N}(0, \sigma^2 I_d)$ (multivariate normal with mean 0 and covariance matrix $I_d$) and a subgaussian matrix $C = (1 - \gamma)I_d + \gamma J_{blocs}$, where $J_{blocs}$ is a bloc matrix such that each $m \times m$ bloc indexed by an action $A_j$ contains only ones and other coefficients are 0. The actions are independent and the sub-arms are correlated with correlation $\gamma$. Then $C^{1/2} = \sqrt{1 - \gamma}I_d + \frac{1}{m}(\sqrt{1 + \gamma(m - 1)} - \sqrt{1 - \gamma})J_{blocs}$. The mean of the best action is taken to be 0 and the mean of other actions is $-\Delta$. The algorithm has access to all these informations, except for the identity of the optimal action.

Suppose for notational convenience that the first action is optimal. The reward of the best action is $\nu_1 = \sqrt{1 + \gamma(m - 1)} \sum_{i \in A_1} \epsilon_i$, while the reward of a sub-optimal action $A_j$ is $\nu_j = \sqrt{1 + \gamma(m - 1)} \sum_{i \in A_j} \epsilon_i - \Delta$.

We use a result from Lai and Robbins [1985] which states that in this bandit case,

$$\liminf_{t \to +\infty} \frac{R_t}{\log t} \geq \sum_{j=2}^{d/m} \frac{\Delta}{D_{KL}(\nu_j, \nu_1)} \, . \tag{2}$$

In our setting, the Kullback-Leibler divergence is

$$
\begin{aligned}
D_{KL}(\nu_j, \nu_1) &= D_{KL}\left(\sqrt{1 + \gamma(m - 1)} \sum_{i \in A_j} \epsilon_i - \Delta, \sqrt{1 + \gamma(m - 1)} \sum_{i \in A_1} \epsilon_i\right) \\
&= D_{KL}\left(\mathcal{N}(-\Delta, \sigma^2 m(1 + \gamma(m - 1))), \mathcal{N}(0, \sigma^2 m(1 + \gamma(m - 1)))\right) \\
&= \frac{\Delta^2}{2\sigma^2 m(1 + \gamma(m - 1))} \, .
\end{aligned}
$$

This together with (2) proves the theorem. $\qquad\square$

## B.2   Upper bound

*Proof.* Lemma 1.

$$
\begin{aligned}
\mathbb{I}\{\overline{\mathbb{G}}_t, \overline{\mathbb{H}}_t\} &= \mathbb{I}\{A^{*\top}\mu \leq A^{*\top}\hat{\mu}_t + E_t(A^*), \overline{\mathbb{H}}_t\} \\
&\leq \mathbb{I}\{A^{*\top}\mu \leq A_t^\top \hat{\mu}_t + E_t(A_t), \overline{\mathbb{H}}_t\} \\
&\leq \mathbb{I}\{A^{*\top}\mu \leq A_t^\top \mu + \frac{\Delta_t}{2} + E_t(A_t)\} \\
&= \mathbb{I}\{\Delta_t \leq 2E_t(A_t)\} \, ,
\end{aligned}
$$

where we used first that the algorithm chooses $A_t = \arg\max_A(A^\top \hat{\mu}_t + E_t(A))$, then that under $\overline{\mathbb{H}}_t$ we have $A_t^\top \hat{\mu}_t \leq A_t^\top \mu + \frac{\Delta_t}{2}$.

$$\mathbb{E}[\sum_{t=1}^T \Delta_t \mathbb{I}\{\overline{\mathbb{G}}_t, \overline{\mathbb{H}}_t\}] \leq \sum_{t=1}^T \mathbb{E}[\Delta_t \mathbb{I}\{\Delta_t \leq 2E_t(A_t)\}] \leq 2\sum_{t=1}^T \mathbb{E}[E_t(A_t)\mathbb{I}\{\Delta_t \leq 2E_t(A_t)\}]$$

$\qquad\square$

*Proof.* Lemma 2.

$$\sum_{t=1}^{T} \Delta_t \mathbb{P}\{\mathbb{H}_t\} \leq \sum_{t=1}^{T} \sum_{i \in A_t} \Delta_t \mathbb{P}\{\mathbb{H}_{i,t}\}$$

$$= \sum_{t=1}^{T} \sum_{i \in A_t} \Delta_t \mathbb{P}\{|\hat{\mu}_t^{(i)} - \mu^{(i)}| \geq \frac{\Delta_t}{2m}\}$$

$$= \sum_{i=1}^{d} \sum_{t=1}^{T} \Delta_t \mathbb{P}\{i \in A_t, |\hat{\mu}_t^{(i)} - \mu^{(i)}| \geq \frac{\Delta_t}{2m}\}$$

$$\leq \Delta_{\max} \sum_{i=1}^{d} \sum_{t=1}^{T} \mathbb{P}\{i \in A_t, |\hat{\mu}_t^{(i)} - \mu^{(i)}| \geq \frac{\Delta_{\min}}{2m}\}$$

$$\leq \Delta_{\max} \sum_{i=1}^{d} \sum_{t=1}^{T} \exp(-\frac{\Delta_{\min}^2 t}{8m^2 C^{(ii)}})$$

$$\leq \frac{8dm^2 \max_i\{C^{(ii)}\} \Delta_{\max}}{\Delta_{\min}^2} .$$

$\square$

*Proof.* Lemma 4.

$$A^{*\top}(\mu - \hat{\mu}_t) = -A^{*\top} D_t^{-1} \sum_{s=1}^{t-1} I_{A_s} C^{1/2} \epsilon_s$$

$$= -A^{*\top} D_t^{-1} \sum_{s=1}^{t-1} I_{A_s \cap A^*} C^{1/2} \epsilon_s$$

$$= -A^{*\top} D_t^{-1} (D + \sum_{s=1}^{t-1} C_{A_s \cap A^*})^{1/2} (D + \sum_{s=1}^{t-1} C_{A_s \cap A^*})^{-1/2} \sum_{s=1}^{t-1} I_{A_s \cap A^*} C^{1/2} \epsilon_s$$

$$\leq \sqrt{A^{*\top} D_t^{-1} (D + \sum_{s=1}^{t-1} C_{A_s \cap A^*}) D_t^{-1} A^*} \left\| \sum_{s=1}^{t-1} I_{A_s \cap A^*} C^{1/2} \epsilon_s \right\|_{(D + \sum_{s=1}^{t-1} C_{A_s \cap A^*})^{-1}}$$

where the last step is the Cauchy-Schwarz inequality. Since $I_{A^*} D \preceq \lambda I_{A^*} \Sigma_C D_t$,

$$A^{*\top}(\mu - \hat{\mu}_t) \leq \sqrt{\lambda A^{*\top} \Sigma_C D_t^{-1} A^* + A^{*\top} D_t^{-1} V_t D_t^{-1} A^*} \sqrt{2 I_{V_t + D}(\epsilon)} .$$

We proved

$$\mathbb{P}\left\{ A^{*\top}(\mu - \hat{\mu}_t) \geq \sqrt{2\tilde{f}(\delta_t) A^{*\top} D_t^{-1} (\lambda \Sigma_C D_t + V_t) D_t^{-1} A^*} \Big| t \in \mathcal{D}_a \right\} \leq \mathbb{P}\left\{ I_{V_t + D}(\epsilon) \geq \tilde{f}(\delta_t) | t \in \mathcal{D}_a \right\} .$$

$\square$

*Proof.* Lemma 5.

Let $\mathcal{F}_t$ be the $\sigma$-algebra $\sigma(A_1, \epsilon_1, \ldots, A_{t-1}, \epsilon_{t-1}, A_t)$. Let $f(u)$ be the density of a multivariate normal random variable independent of all other variables with mean 0 and covariance $D^{-1}$. Then from the proof of lemma 9 of Abbasi-Yadkori et al. [2011],

$$M_t(D) = \sqrt{\frac{\det D}{\det(D + V_t)}} \exp(I_{V_t + D}(\epsilon))$$

$$= \int_{\mathbb{R}^d} \exp\left( u^\top \sum_{s=1}^{t-1} I_{A_s \cap A^*} C^{1/2} \epsilon_s - \frac{1}{2} u^\top \sum_{s=1}^{t-1} C_{A_s \cap A^*} u \right) f(u) du$$

We define the following quantities, for $u \in \mathbb{R}^d$,

$$M_t^u = \exp\left( u^\top \sum_{s=1}^{t-1} I_{A_s \cap A^*} C^{1/2} \epsilon_s - \frac{1}{2} u^\top \sum_{s=1}^{t-1} C_{A_s \cap A^*} u \right),$$

$$D_s^u = \exp\left( u^\top I_{A_s \cap A^*} C^{1/2} \epsilon_s - \frac{1}{2} u^\top C_{A_s \cap A^*} u \right)$$

such that $M_t^u = \prod_{s=1}^{t-1} D_s^u$.

From the subgaussian property of $\epsilon_s$, $\mathbb{E}[D_s^u | \mathcal{F}_s] \leq 1$.

$$
\begin{aligned}
\mathbb{E}[M_t^u | \mathcal{F}_{s-1}] &= \mathbb{E}[\prod_{s=1}^{t-1} D_s^u | \mathcal{F}_{s-1}] \\
&= (\prod_{s=1}^{t-2} D_s^u) \mathbb{E}[D_{t-1}^u | \mathcal{F}_{t-1}] \\
&\leq M_{t-1}^u .
\end{aligned}
$$

Thus $M_t^u$ is a supermartingale and $\mathbb{E}[M_1^u] = \mathbb{E}[D_1^u] \leq 1$. So for all $t$, $\mathbb{E}[M_t^u] \leq 1$.

Finally, $\mathbb{E}[M_t(D)] = \mathbb{E}_u \mathbb{E}[M_t^u | u] \leq 1$. $\qquad \square$

*Proof.* Lemma 6.

We have $\frac{1}{1+\eta} I_{A^*} D_t \preceq D_a \preceq I_{A^*} D_t$. We use $D = \lambda \Sigma_C D_a + I_{[d] \setminus A^*}$. Then $\frac{\lambda}{1+\eta} I_{A^*} \Sigma_C D_t \preceq D$. The $I_{[d] \setminus A^*}$ part in $D$ is there only to satisfy the positive definiteness and has no consequence.

These matrix inequalities show that $D^{-1/2} V_t D^{-1/2} \preceq \frac{1+\eta}{\lambda} D_t^{-1/2} \Sigma_C^{-1/2} V_t \Sigma_C^{-1/2} D_t^{-1/2}$, which is $\frac{1+\eta}{\lambda}$ times a matrix with $m$ ones and $d - m$ zeros on the diagonal. The determinant of a positive definite matrix is smaller than the product of its diagonal terms, so

$$
\begin{aligned}
\det(I_d + D^{-1/2} V_t D^{-1/2}) &\leq \det(I_d + \frac{1+\eta}{\lambda} D_t^{-1/2} \Sigma_C^{-1/2} V_t \Sigma_C^{-1/2} D_t^{-1/2}) \\
&\leq (1 + \frac{1+\eta}{\lambda})^m .
\end{aligned}
$$

$\qquad \square$

*Proof.* Lemma 7.

Let $j_0$ such that $\beta_{j_0, e} \leq 1/m$. Then for all $j > j_0$,

$$\mathbb{A}_{t,e}^j = \{|S_{t,e}^j| \geq 1; \forall k < j_0, |S_{t,e}^k| < m\beta_{k,e}; \forall k \in \{j_0, \ldots, j-1\}, |S_{t,e}^k| = 0\} .$$

But as the sequence of sets $(S_{t,e}^j)_j$ is decreasing, $\{|S_{t,e}^{j_0}| = 0\}$ and $\{|S_{t,e}^{j_0}| \geq 1\}$ cannot happen simultaneously. $\mathbb{A}_{t,e}^j$ cannot happen for $j > j_0$. $\qquad \square$

*Proof.* Lemma 8.

First we rewrite $\overline{\mathbb{A}_{t,1}}$, following Kveton et al. [2015],

$$
\begin{aligned}
\overline{\mathbb{A}_{t,1}} &= \cap_{j=1}^{j_0} \overline{\mathbb{A}_{t,1}^j} \\
&= \cap_{j=1}^{j_0} \left[ \{|S_{t,1}^j| < m\beta_{j,1}\} \cup (\cup_{k=1}^{j-1} \{|S_{t,1}^k| \geq m\beta_{k,1}\}) \right] \\
&= \cap_{j=1}^{j_0} \{|S_{t,1}^j| < m\beta_{j,1}\} \\
&= (\cap_{j=1}^{j_0-1} \{|S_{t,1}^j| < m\beta_{j,1}\}) \cap \{|S_{t,1}^{j_0}| = 0\} .
\end{aligned}
$$

The complementary set of $S_{t,1}^j$ in $A_t$ is $\overline{S}_{t,1}^j = \{i \in A_t, i \notin S_{t,1}^j\}$. If we have $\overline{\mathbb{A}_{t,1}}$, then $\overline{S}_{t,1}^{j_0} = A_t$ and thus

$$\sum_{i \in A_t} \frac{\Gamma^{(ii)}}{n_t^{(i)}} = \sum_{j=1}^{j_0} \sum_{i \in \overline{S}_{t,1}^j \setminus \overline{S}_{t,1}^{j-1}} \frac{\Gamma^{(ii)}}{n_t^{(i)}}$$

$$< \sum_{j=1}^{j_0} \sum_{i \in \overline{S}_{t,1}^j \setminus \overline{S}_{t,1}^{j-1}} \frac{\Delta_t^2}{8 f(t) g_1(m, \gamma_t) \alpha_{j,1}}$$

$$= \frac{\Delta_t^2}{8 f(t) g_1(m, \gamma_t)} \sum_{j=1}^{j_0} \frac{|\overline{S}_{t,1}^j \setminus \overline{S}_{t,1}^{j-1}|}{\alpha_{j,1}}$$

$$< \frac{m \Delta_t^2}{8 f(t) g_1(m, \gamma_t)} \left( \sum_{j=1}^{j_0} \frac{\beta_{j-1,1} - \beta_{j,1}}{\alpha_{j,1}} + \frac{\beta_{j_0,1}}{\alpha_{j_0,1}} \right), \tag{3}$$

where (3) follows the same steps as lemma 4 of Kveton et al. [2015].

Proof of the second statement of the lemma:

First, follow the same steps as before to get

$$\sum_{i \in A_t} \sqrt{\frac{\Gamma^{(ii)}}{n_t^{(i)}}} < \frac{m \Delta_t}{\sqrt{8 f(t) g_2(m, \gamma_t)}} \left( \sum_{j=1}^{j_0} \frac{\beta_{j-1,2} - \beta_{j,2}}{\sqrt{\alpha_{j,2}}} + \frac{\beta_{j_0,2}}{\sqrt{\alpha_{j_0,2}}} \right).$$

Then take the limit when $j_0 \to +\infty$. $\qquad\square$

*Proof.* Lemma 10.

We break the events $\mathbb{A}_{t,e}^j$ into sub-events $\mathbb{A}_{t,e}^{j,a} = \mathbb{A}_{t,e}^j \cap \{a \in A_t, n_t^{(a)} \leq \alpha_{j,e} \frac{8 f(t) \Gamma^{(aa)} g_e(m, \gamma_t)}{\Delta_t^2}\}$ that at least $m\beta_{j,e}$ arms are not pulled often and that the arm $a$ is one of them. Since $\mathbb{A}_{t,e}^j$ implies that at least $\beta_{j,e} m$ arms are pulled less than the threshold, we have

$$\mathbb{I}\{\mathbb{A}_{t,e}^j\} \leq \frac{1}{m\beta_{j,e}} \sum_{a=1}^d \mathbb{I}\{\mathbb{A}_{t,e}^{j,a}\}.$$

The regret that we want to bound is

$$\sum_{t=1}^T \Delta_t \mathbb{I}\{\overline{\mathbb{H}}_t \cap \overline{\mathbb{G}}_t\} \leq \sum_{e=1}^2 \sum_{t=1}^T \sum_{j=1}^{j_0} \Delta_t \mathbb{I}\{\mathbb{A}_{t,e}^j\}$$

$$\leq \sum_{e=1}^2 \sum_{t=1}^T \sum_{j=1}^{j_0} \sum_{a=1}^d \frac{\Delta_t}{m\beta_{j,e}} \mathbb{I}\{\mathbb{A}_{t,e}^{j,a}\}.$$

Each arm $a$ is contained in $N_a$ actions. Let $\Delta_{a,1} \geq \ldots \geq \Delta_{a,N_a}$ be the gaps of these actions and let $\Delta_{a,0} = +\infty$. Then

$$\sum_{t=1}^T \Delta_t \mathbb{I}\{\overline{\mathbb{H}}_t \cap \overline{\mathbb{G}}_t\} \leq \sum_{e=1}^2 \sum_{t=1}^T \sum_{j=1}^{j_0} \sum_{a=1}^d \sum_{n=1}^{N_a} \frac{\Delta_{a,n}}{m\beta_{j,e}} \mathbb{I}\{\mathbb{A}_{t,e}^{j,a}, \Delta_t = \Delta_{a,n}\}$$

$$\leq \sum_{e=1}^2 \sum_{t=1}^T \sum_{j=1}^{j_0} \sum_{a=1}^d \sum_{n=1}^{N_a} \frac{\Delta_{a,n}}{m\beta_{j,e}} \mathbb{I}\{a \in A_t, n_t^{(a)} \leq \alpha_{j,e} \frac{8 f(t) \Gamma^{(aa)} g_e(m, \gamma_t)}{\Delta_{a,n}^2}, \Delta_t = \Delta_{a,n}\}$$

Let $\theta_{j,e,a} = \alpha_{j,e} 8f(T)\Gamma^{(aa)}g_e(m,\gamma)$. In the following equations, changes between successive lines are highlighted in blue.

$$\sum_{t=1}^{T}\sum_{n=1}^{N_a}\Delta_{a,n}\mathbb{I}\{a \in A_t, n_t^{(a)} \leq \alpha_{j,e}\frac{8f(t)\Gamma^{(aa)}g_e(m,\gamma_t)}{\Delta_{a,n}^2}, \Delta_t = \Delta_{a,n}\}$$

$$\leq \sum_{t=1}^{T}\sum_{n=1}^{N_a}\Delta_{a,n}\mathbb{I}\{a \in A_t, n_t^{(a)} \leq \frac{\theta_{j,e,a}}{\Delta_{a,n}^2}, \Delta_t = \Delta_{a,n}\}$$

$$= \sum_{t=1}^{T}\sum_{n=1}^{N_a}\sum_{p=1}^{n}\Delta_{a,n}\mathbb{I}\{a \in A_t, n_t^{(a)} \in (\frac{\theta_{j,e,a}}{\Delta_{a,p-1}^2}, \frac{\theta_{j,e,a}}{\Delta_{a,p}^2}], \Delta_t = \Delta_{a,n}\}$$

$$\leq \sum_{t=1}^{T}\sum_{n=1}^{N_a}\sum_{p=1}^{n}\Delta_{a,p}\mathbb{I}\{a \in A_t, n_t^{(a)} \in (\frac{\theta_{j,e,a}}{\Delta_{a,p-1}^2}, \frac{\theta_{j,e,a}}{\Delta_{a,p}^2}], \Delta_t = \Delta_{a,n}\}$$

$$\leq \sum_{t=1}^{T}\sum_{n=1}^{N_a}\sum_{p=1}^{N_a}\Delta_{a,p}\mathbb{I}\{a \in A_t, n_t^{(a)} \in (\frac{\theta_{j,e,a}}{\Delta_{a,p-1}^2}, \frac{\theta_{j,e,a}}{\Delta_{a,p}^2}], \Delta_t = \Delta_{a,n}\}$$

$$\leq \sum_{t=1}^{T}\sum_{p=1}^{N_a}\Delta_{a,p}\mathbb{I}\{a \in A_t, n_t^{(a)} \in (\frac{\theta_{j,e,a}}{\Delta_{a,p-1}^2}, \frac{\theta_{j,e,a}}{\Delta_{a,p}^2}], \Delta_t > 0\}$$

$$\leq \frac{\theta_{j,e,a}}{\Delta_{a,1}} + \sum_{p=2}^{N_a}\theta_{j,e,a}\Delta_{a,p}(\frac{1}{\Delta_{a,p}^2} - \frac{1}{\Delta_{a,p-1}^2})$$

$$\leq \frac{\theta_{j,e,a}}{\Delta_{a,1}} + \theta_{j,e,a}\int_{\Delta_{a,N_a}}^{\Delta_{a,1}} x^{-2}dx$$

$$= \frac{\theta_{j,e,a}}{\Delta_{a,N_a}} = \frac{\theta_{j,e,a}}{\Delta_{a,\min}}.$$

$$\sum_{t=1}^{T}\Delta_t\mathbb{I}\{\overline{\mathbb{H}}_t \cap \overline{\mathbb{G}}_t\} \leq \sum_{e=1}^{2}\sum_{a\in[d]}\sum_{j=1}^{j_0}\frac{\theta_{j,e,a}}{m\beta_{j,e}\Delta_{a,\min}}$$

$$= 8f(T)\sum_{a\in[d]}\frac{\Gamma^{(aa)}}{\Delta_{a,\min}}\sum_{e=1}^{2}\frac{g_e(m,\gamma)}{m}\left(\sum_{j=1}^{j_0}\frac{\alpha_{j,e}}{\beta_{j,e}}\right)$$

$$= 16f(T)\sum_{a\in[d]}\frac{\Gamma^{(aa)}}{\Delta_{a,\min}}\left[(\lambda+1-\gamma)l_1\sum_{j=1}^{j_0}\frac{\alpha_{j,1}}{\beta_{j,1}} + \gamma m l_2^2\sum_{j=1}^{j_0}\frac{\alpha_{j,2}}{\beta_{j,2}}\right].$$

$\square$

## C Finding the best sequences for the sums indexed by 1.

The constraints on the four sequences $(\alpha_{i,1})_{i\geq1}$, $(\alpha_{i,2})_{i\geq1}$, $(\beta_{i,1})_{i\geq0}$ and $(\beta_{i,2})_{i\geq0}$ are that they must be positive decreasing with limit 0, $\beta_{0,1} = \beta_{0,2} = 1$ and $\lim_{j\to+\infty}\beta_{j,2}/\sqrt{\alpha_{j,2}} = 0$.

For $i \geq 1$, we take $\beta_{i,1} = \alpha_{i,1} = \beta^i$ with $\beta \in (0,1)$. Then $j_{0,1} = \lceil\frac{\log m}{\log 1/\beta}\rceil$ and $l_1\sum_{j=1}^{j_{0,1}}\frac{\alpha_{j,1}}{\beta_{j,1}} = j_{0,1}^2(1/\beta - 1) + j_{0,1} \leq j_{0,1}^2/\beta$. We take $\beta_{i,2}$ and $\alpha_{i,2}$ as in Kveton et al. [2015]: $\beta_{i,2} = \beta_2^i$ with $\beta_2 = 0.236$; $\alpha_{i,2} = \left(\frac{1-\beta_2}{\sqrt{\alpha}-\beta_2}\right)^2\alpha^i$, with $\alpha = 0.146$. Then $\sum_{j=1}^{+\infty}\frac{\alpha_{j,2}}{\beta_{j,2}} \leq 45$ and $l_2 \leq 1$.

The optimal choice for $\beta$ is close to $1/5$, value for which we get the wanted regret bound.

We now show that the choice of $\alpha_{j,1}$ and $\beta_{j,1}$ made previously is close to optimal among the sequences with $\alpha_{j,1} = \beta_{j,1}$.

**Lemma 11.** *Suppose that for all $j$, $\alpha_j = \beta_j$. Then the optimal value $v$ of $(\sum_{j=1}^{j_0} \frac{\alpha_j}{\beta_j})(\frac{\beta_{j_0}}{\alpha_{j_0}} + \sum_{j=1}^{j_0} \frac{\beta_{j-1}-\beta_j}{\alpha_j})$ is such that*

$$v \geq 1.54 \log^2 m .$$

*Proof.* We want to minimize $j_0 \sum_{j=1}^{j_0} (\frac{\beta_{j-1}}{\beta_j} - 1)$ under the constraint that $j_0 = \min\{j : \beta_j \leq \frac{1}{m}\}$. Let $h_j = \frac{\beta_{j-1}}{\beta_j} - 1$, then the decreasing constraint on the $(\beta_j)$ sequence imposes that for all $j \geq 1$, $h_j \geq 0$. Also, $\beta_{j_0} \leq \frac{1}{m}$ implies $\prod_{j=1}^{j_0} \frac{1}{h_j+1} \leq \frac{1}{m}$.

We solve the following minimization problem:

$$\text{minimize over } j_0, (h_j): \ j_0(1 + \sum_{j=1}^{j_0} h_j)$$

$$\text{such that: } \forall j \geq 1, h_j \geq 0 ,$$

$$\prod_{j=1}^{j_0} (h_j + 1) \geq m ,$$

$$j_0 \geq 1 .$$

For a fixed $j_0$, the minimum in $(h_j)$ is attained for all $h_j$ equal to $m^{1/j_0} - 1$. Then we should minimize $j_0^2(m^{1/j_0} - 1) + j_0$ with respect to $j_0$. We will instead write that $j_0^2(m^{1/j_0} - 1) + j_0 \geq j_0^2(m^{1/j_0} - 1)$ and find a lower bound with this latter expression.

Let $f(x) = x^2(m^{1/x} - 1)$ for $x \in \mathbb{R}^+$. Then with $y = \frac{x}{\log m}$, $f(x) = y^2(e^{1/y} - 1)\log^2 m$. The function $g(y) = y^2(e^{1/y} - 1)$ has a unique minimum and plotting $g$ shows that this minimum $x^*$ is such that $f(x^*) \geq 1.54 \log^2 m$. $\qquad\square$