[Reviews · NeurIPS 2016]

Reviewer 1

Summary

This paper studies a online linear optimization problem with combinatorial action sets and 'semi-bandit' feedback. At any given time, the reward noise of individual arms is correlated. The paper provides a regret bound for a linear UCB algorithm that scales with the degree of correlation and interpolates between two extreme cases studied in the prior literature: the case with no correlation, and the case where observations on different arms may be perfectly correlated.

Qualitative Assessment

This paper makes an interesting contribution. It's natural for there be correlation between arm outcomes, e.g. the realized time to traverse neighboring edges on a graph, and it's valuable to pin down the dependence of regret bounds on this correlation. This result helps link and unify separate results in the literature. There are few limitations to the results. First, the proposed algorithm requires solving an optimization problem that can be NP hard. It's not clear the algorithm is implementable. Second, it seems this UCB approach may be statistically suboptimal in some settings. When possible, the most efficient way to gather information would be to select base arms with relatively uncorrelated outcomes. The algorithm won't do this. Finally, the regret bounds depend on the inverse mean-gaps that can be incredibly small. (e.g. the difference between the shortest and second shortest path in a graph) Gap-independent results would be nice here. -- Here are some additional comments--- • It would be nice to provide a better motivation for the problem, and better intuition for the results. For example, it's very natural that one attains m times more information with no correlation than under perfect correlation. • Do you need to depend on the largest correlation coefficient? Presumably, the reward of two base arms could be perfectly correlated without performance degrading substantially. The issue arises when there is a great deal of correlation on average. • The proof looks familiar from other UCB analyses, but I find it very hard to read. I'm not sure this is necessary. • The lower bound statement is not precise. It's not clear precisely what it means that "there exists a problem with gaps Delta" (all gaps = Delta? Is a gap a formal term? ). Also arms cannot be sub-Gaussian, you mean that their reward noise is sub-Gaussian.

Confidence in this Review

2-Confident (read it all; understood it all reasonably well)


Reviewer 2

Summary

The paper looks at the (stochastic) combinatorial semi-bandit. Here the decision set is some subset of {0, 1}^d. At each round the action picked is A_t \in {0, 1}^d. The feedback revelaed is p_i for each i such that $i$\th element of A_t is 1 (thus, semi-bandit). However, these p_i may not necessarily be independent. The present paper extrapolates between the worst-case (no assumption) or independence assumptions in existing work. The same algorithm essentially gives optimal regret (up to poly log m factors) in both settings.

Qualitative Assessment

Overall the paper is quite well-written. The arguments are similar to those appearing in most such papers, and as such are not particularly novel or technically interesting. Nevertheless, the paper tells a good story. One point that is not pointed out, is the requirement that exactly *m* items of A_t picked at time $t$ should be 1. Otherwise, I'm not sure the tightness will be achieved up to poly log m factors. It would be good to mention this explicitly. Other comments: - (Line 36): $\Delta_{i, min}$ - is not great as notation, given that you already have $\Delta_t$ - (Line 56): $\Delta$ is never defined.

Confidence in this Review

2-Confident (read it all; understood it all reasonably well)


Reviewer 3

Summary

This paper studies combinatorial semi-bandit with known covariance. It proposes OLS-UCB, a UCB-like algorithm for the considered problem (Algorithm 1), and establishes both gap-dependent and gap-free regret bounds (Theorem 2 and Corollary 1). The derived regret bounds show the transition between a general case and an independent case.

Qualitative Assessment

This paper is interesting and well-written in general. To the best of my knowledge, the analyses in this paper are also technically correct. My major concern about this paper is its novelty. Based on existing literature (especially Kveton et al. 2015 and Combes et al. 2015), both the proposal of OLS-UCB and its analyses are somewhat straightforward. My understanding is that NIPS papers should have notable novel contributions, and I do not think this paper has met this standard. Moreover, this paper does not have any experiment results. The paper will be much stronger if the authors can show that their proposed algorithm outperforms state-of-the-art combinatorial semi-bandit algorithms in representative practical problems. That said, I vote for a weak reject of this paper.

Confidence in this Review

3-Expert (read the paper in detail, know the area, quite certain of my opinion)


Reviewer 4

Summary

This paper deals with the stochastic combinatorial semi-bandit problem in the setting where the joint reward vector is subgaussian (after subtracting its mean) and an "upper bound" (in some sense) of the subgaussian covariance matrix is known as a prior information. An algorithm OLS-UCB is provided, which combines linear regression and UCB, and both gap-dependent and gap-independent regret upper bounds are shown. The authors also prove an asymptotic gap-dependent regret lower bound, matching their gap-dependent upper bound (up to a polylogarithmic factor) when all actions have the same gap and all arms have the same variance upper bound.

Qualitative Assessment

The authors consider an interesting direction of the stochastic combinatorial semi-bandit problem. The subgaussian setting subsumes the case of bounded rewards studied in most bandit literatures, and an O(log T) gap-dependent bound and an Õ(\sqrt{T}) gap-independent bound on the expected regret are provided. However, the paper has several drawbacks: - The algorithm is not computationally efficient. This is also mentioned in the paper. - The gap-independent bound (Corollary 1) is only stated for the case of equal gaps. Can it be generalized to the general case? - It is unclear how OLS-UCB performs compared with other existing algorithms (such as CombUCB1 in [Kveton et al., AISTATS'2015]) in practice. It would be better to have an experimental comparison. - The paper is not very well written in general. There are numerous typos/English mistakes. There are also some minor mistakes in the statement of Theorem 2: the summation over {i \in [d]} should be replaced by {i \in [d], \Delta_{i,\min} > 0}; \Delta_{\min} and \Delta_{\max} are undefined.

Confidence in this Review

1-Less confident (might not have understood significant parts)


Reviewer 5

Summary

This work studies the stochastic combinatorial semi-bandit problem. The main contribution of this work is to analyze how to exploit the dependency structure between the different arms. In particular, the authors assume that the arms are subgaussian and that the algorithm has some prior knowledge of the subgaussian constants. Under this assumption the authors present the following 1) A lowerbound on the regret of any algorithm that depends on this prior knowledge 2) An algorithm OLS-UCB based on ideas from the linear bandit setting that employs this prior knowledge. 3) Analysis of the regret of this algorithm that shows a nice transition from the general case (no assumption made on the relationship between arms) to the independent case. 4) In the independent case the OLS-UCB algorithm is the same as the ESCB-2 algorithm of Combes et al. The authors have an improved analysis of the regret bound on the ESCB-2 algorithm. 5) In the general case the regret bound of OLS-UCB matches the regret bound of Kveton et al but suffers from the problem that it is computationally inefficient. The algorithm of Kveton et al can be solved using an LP, but OLS-UCB computes an argmax over the actions.

Qualitative Assessment

This is a clearly written and well motivated work where the authors study the stochastic combinatorial semi-bandit problem and generalize prior work in this area. This fits very nicely with the existing literature in this area and shows a smooth transition between the general setting and the independent setting. To the best of my knowledge, this is a novel contribution and raises a few interesting questions and ideas about future work in this area. My main criticism is that the algorithm is impractical in a few ways (as the authors themselves acknowledge) 1) It is not clear how one should go about estimating the subgaussian covariance matrix. Framing the problem in this way (prior knowledge of subgaussian constants) it is not *very* surprising that using a linear regression is the way to go, since it is the combinatorial analogue of the fixed design setting. Thus the "prior knowledge setting" looks somewhat contrived. 2) Computing the argmax over the actions. In this respect the algorithm is similar to the ESCB algorithm, however the authors perform an improved regret analysis.

Confidence in this Review

2-Confident (read it all; understood it all reasonably well)


Reviewer 6

Summary

The submitted paper considers combinatorial semi-bandit stochastic semi-bandit problem. A brief description of this problem is as follows: Combinatorial means that we, at each stage, selects a subset of arms to pull. Semi-bandit means that we see the rewards of all pulled arms. Finally, stochastic means that the rewards are drawn from some unknown distributions that may be correlated. The difficulty of this problem heavily relies on the (unknown) correlations between the arms. (If they are independent it is easier.) Previous work has developed different algorithms to deal with the extreme cases. The goal of the paper is to present one algorithm that works well in all cases, i.e., interpolate between the easy to hard case. I would say that the paper does progress towards this goal but doesn't quite reach it. This is because their algorithm heavily relies on being given an estimate of the correlation matrix (the matrix Gamma). As the authors write in the conclusions, it would indeed be interesting to estimate this covariance matrix online to get good regret bounds without prior knowledge.

Qualitative Assessment

The problem addressed in the paper is interesting and will most likely generate quite some interest. The reasons for not being more excited are: - Need an estimation of the covariance matrix up front. - As a non-expert, the techniques seem to heavily build on previous works, i.e., it is unclear what the technical insights are. Overall, I think it is a nice submission.

Confidence in this Review

1-Less confident (might not have understood significant parts)